# Oligodendroglial Heterogeneity in Neuropsychiatric Disease

**DOI:** 10.3390/life11020125

**Published:** 2021-02-06

**Authors:** Sunniva M. K. Bøstrand, Anna Williams

**Affiliations:** Centre for Regenerative Medicine, Institute for Regeneration and Repair, The University of Edinburgh, Edinburgh BioQuarter, 5, Little France Drive, Edinburgh EH16 4UU, UK; sunniva.bostrand@ed.ac.uk

**Keywords:** oligodendrocyte, neuropsychiatric disease, heterogeneity, Huntington’s disease

## Abstract

Oligodendroglia interact with neurons to support their health and maintain the normal functioning of the central nervous system (CNS). Human oligodendroglia are a highly heterogeneous population characterised by distinct developmental origins and regional differences, as well as variation in cellular states, as evidenced by recent analysis at single-nuclei resolution. Increasingly, there is evidence to suggest that the highly heterogeneous nature of oligodendroglia might underpin their role in a range of CNS disorders, including those with neuropsychiatric symptoms. Understanding the role of oligodendroglial heterogeneity in this group of disorders might pave the way for novel approaches to identify biomarkers and develop treatments.

## 1. Introduction

Oligodendroglia interact with neurons to maintain the health and normal functioning of the central nervous system (CNS) by supporting axonal health and facilitating the rapid transmission of action potentials by saltatory conduction. The oligodendroglial lineage includes the self-renewing oligodendrocyte precursor cells (OPCs) and their progeny oligodendrocytes. In addition to enabling rapid conduction of action potentials, myelination by oligodendrocytes plays an important role in providing metabolic support for axons, and impairment to this function has been shown to contribute to non-cell autonomous neurodegeneration [1,2,3,4], demonstrating that changes to oligodendroglia may increase neuronal vulnerability. Consequently, as oligodendroglia are necessary to support the healthy functioning of the CNS, damage to these cell types or the myelin they produce is a common feature in many neurological disorders. Most prominently, the demyelinating disorders Multiple Sclerosis (MS) and many leukodystrophies are characterised by demyelination and white matter (WM) lesions, giving rise to neurological impairments in development and adulthood. In addition to these kinds of disorders where myelin changes and WM damage are recognised as the core pathology, there is an increasing number of CNS disorders where the role of WM damage is becoming more apparent, including a wide range of neuropsychiatric conditions. The observation of macroscopic changes to WM in patients raises the question of whether oligodendroglia are dysfunctional in these conditions and whether manipulating oligodendroglia therapeutically might improve the pathology and clinical course of the disease. Myelin and oligodendroglial changes in neuropsychiatric disease have been studied using a number of different approaches, including magnetic resonance imaging (MRI)-based method diffusion tensor imaging (DTI) used to assess the structural integrity of WM tracts in patients as compared to healthy individuals. In addition, oligodendroglial dysfunction and myelin integrity have been studied in post mortem (PM) tissue from patients and animal models at a histopathological and ultrastructural level, respectively. Lastly, novel approaches to high-throughput transcriptomic analysis at the level of single cells and nuclei enable the characterisation of oligodendroglia and other CNS cell types at an unprecedented level of detail, and facilitates the investigation of whether altered heterogeneity of these cell populations may constitute part of the pathology of neuropsychiatric conditions. In this review, we consider the evidence for the involvement of oligodendroglia in a range of neuropsychiatric disorders where this has been less recognised previously, with a focus particularly on the role of oligodendroglial heterogeneity.

## 2. White Matter Damage and Oligodendroglial Dysfunction in Neuropsychiatric Conditions

Although the majority of neuropsychiatric conditions are traditionally thought to arise as a result of neuronal impairment, there is increasing evidence that oligodendroglial dysfunction plays a role in a number of such conditions. This has been found both in classical dementias such as Huntington’s (HD) [5], Alzheimer’s (AD) [6] and Parkinson’s Disease (PD) [7,8], as well as in more typical psychiatric conditions like Schizophrenia (Sz) [9], Major Depressive Disorder (MDD) [10], and Alcohol Use Disorder (AUD) [11]. Across a number of these disorders, macroscopic changes to WM have been observed in patients using MRI techniques such as DTI [8,9,12] to obtain measures such as fractional anisotropy, which reflects the directionality of diffusing water molecules and has been shown to correlate strongly with myelination [13], as well as changes to oligodendroglia at the molecular and cellular level [5,6,10,11,14]. This oligodendroglial involvement has been studied relatively extensively in HD, and so here we will illustrate this first focusing on this disease.

### 2.1. The Role of Oligodendroglia in Huntington’s Disease

Huntington’s disease serves as a prominent example of a condition with a known cause where neuronal loss is the core pathology, but there is accumulating evidence for a role of oligodendroglial dysfunction in the disease. HD is a rare, heritable condition caused by a CAG trinucleotide repeat expansion in the Huntingtin (HTT) gene, leading to fatal neurodegeneration which initially affects the medium spiny neurons of the striatum. While neuronal cell death is considered the primary pathology of HD, WM deficits and changes to oligodendroglia are also typical of the disease [5,15,16,17]. Global changes to WM microstructure are common in patients with HD, and these have been shown to associate with the core symptoms of the disease which includes motor deficits, cognitive and psychiatric problems [18,19,20]. WM changes are also observed in premanifest HD, i.e., individuals who are gene expansion-positive but have yet to present with any of the core HD symptoms. Several DTI studies have demonstrated alterations to WM microstructure at this stage [21,22,23,24], suggesting that changes to WM represent an early feature of the disease that manifests prior to symptom onset. This in turn hints at the possibility that oligodendroglial pathology in HD might precede neuronal loss.

Additional evidence of early WM pathology and oligodendroglial dysfunction onset prior to overt neuronal death has been shown in several mouse models of HD [25,26], and different transgenic animal models have been used to investigate the effect of the expanded mutant *HTT* (*mHTT*) expression more specifically in oligodendroglial populations. In one example, Huang et al. [27] expressed a *HTT* fragment under the oligodendrocyte promoter PLP, and compared the effects of a fragment with expanded CAG repeat length to a healthy *HTT* fragment. Animals expressing the expanded *mHTT* only in oligodendrocytes displayed a typical HD phenotype characterised by motor deficits alongside progressive weight loss and premature death. These animals also showed decreased expression of mature oligodendrocyte markers such as myelin basic protein (MBP) and myelin-oligodendrocyte glycoprotein (MOG), as well as thinning of the myelin sheaths and shortened oligodendrocyte processes. Ferrari-Bardile et al. [26] applied the reverse approach, of lowering the expression of *mHTT* under the glial progenitor promoter NG2, thus reducing levels of *mHTT* only in oligodendroglia, using a different transgenic mouse model of HD expressing *mHTT* globally. This was sufficient to rescue myelin deficits in the HD mouse, as well as improving performance in behavioural tests. These studies show that targeting *mHTT* expression in oligodendroglia has a prominent effect on both the cellular and behavioural phenotype in two different mouse models of HD, suggesting that early oligodendrocyte dysfunction in HD has an independent contribution to disease pathology that is not simply secondary to neuronal loss.

### 2.2. The Role of Oligodendroglia in Other Neuropsychiatric Diseases

HD is not the only example of a neuropsychiatric disease where oligodendroglial pathology may be important as an early finding and might be driving some of the pathology and clinical phenotype. Such early WM and oligodendroglial changes as observed in HD have also been reported in other conditions. In Sz, disruptions to WM integrity correlates with cognitive impairment, and oligodendrocyte gene expression is commonly dysregulated [9]. Hakak et al. [28] found transcriptional downregulation of genes which are enriched in myelin-forming oligodendrocytes, such as myelin-associated glycoprotein (MAG), and 2′, 3′-Cyclic-nucleotide 3′-phosphodiesterase (CNPase) in patients with Sz. Impairment to the WM structure of the fornix affecting the hippocampal-prefrontal cortex (PFC) functional circuit has been shown to correlate with impaired neurocognitive function in MDD [12]. Similarly, in early PD patients, Duncan et al. [8] observed that increased mean diffusivity as measured using DTI, reflecting impaired WM microstructure, in frontal and parietal tracts corresponded to impaired semantic and executive functioning. Furthermore, recent studies of the human PM brain using transcriptomic profiling at single-nuclear resolution implicate a novel role of oligodendroglia in these conditions [7,10], as discussed in greater detail below. These results showing alterations to WM structural and functional connectivity, and oligodendroglial transcriptomic profiles across different disease states suggest that this is common to diverse diseases, and raises the hypothesis that there may be a similar underlying pathological process.

## 3. Heterogeneity of the Oligodendroglial Lineage

The involvement of oligodendroglia in diverse neuropsychiatric and neurodegenerative diseases poses the question of whether a disease-associated oligodendroglial phenotype exists. In order to formulate an answer to this question, we must first consider that the oligodendroglial lineage consists of highly heterogeneous phenotypes. The heterogeneity of oligodendroglia in the CNS has been recognised since they were first categorised through detailed histological observations in terms of their morphological and regional diversity by Pio Rio del Hortega in 1928 [29]. This initial characterisation of different oligodendroglial subtypes proposed important ideas about their heterogeneity that are still prominent today (including the suggestion that oligodendroglia are heterogeneous in terms of the myelin sheath lengths they produce). Heterogeneity of oligodendroglia arises from various sources (Figure 1), and is reflected in their diverse developmental origins [30], the varying potential of OPCs to differentiate [31] and proliferate [32], and the propensity for mature oligodendrocytes to produce myelin sheaths of a certain number [33] and length [34], as well as their ability to remyelinate following injury [35]. For the purpose of the present review, we will consider heterogeneity as an inclusive term which captures any dimensions along which oligodendroglia can vary, and some of these are discussed below.

### 3.1. Developmental Heterogeneity

Developmental origin represents an important aspect of oligodendroglial heterogeneity. In the developing forebrain of the rodent, oligodendrocytes arise in three temporally distinct waves [36]. The first, starting at embryonic day (E) 11.5, arises from Nkx2.1-expressing precursors in the medial ganglionic eminence and anterior entopeduncular area of the developing forebrain. A second wave of Gsh2-positive oligodendroglia at E15 arises from the lateral and caudal ganglionic eminence with a third and final, postnatal wave of dorsal origin outcompeting the previously generated oligodendroglia. Using an induced pluripotent stem cell (iPSC)-derived organoid model, human oligodendroglia have also been shown to arise from both ventral and dorsal origins in vitro [39]. As in rodents, ventral oligodendroglia develop prior to those of dorsal origin and when fusing ventral and dorsal organoids, the later-born dorsal oligodendroglia outcompete the ventral, much like dorsal oligodendroglia dominate in the postnatal murine brain. Furthermore, Crawford et al. [35] have shown that the developmental origin of oligodendroglia has an impact on their remyelinating potential in the context of injury and ageing. Using a dual-reporter mouse line in order to differentially label oligodendroglia of dorsal and ventral origin, they investigated the ability of OPCs to differentiate and remyelinate following injury in neonatal and aged animals. Their results showed that following demyelination of the corpus callosum in young animals, dorsal OPCs dominated the remyelination response with ventral OPCs lacking the same propensity for regeneration. Subsequent experiments found the same effect in aged animals, and showed that the differences were more pronounced in the adult as compared to the neonatal brain, which may reflect a differential susceptibility of the ventral and dorsal populations to ageing. By culturing ventral and dorsal OPCs under identical conditions in vitro, the authors were able to demonstrate that the differences observed between OPCs of dorsal and ventral origin are intrinsic, rather than being determined by the external environment. These studies demonstrate prominent developmental heterogeneity of oligodendroglia, with similarities observed between human and mouse. Furthermore, they show that developmental origin determines intrinsic functional properties of OPCs, both early in life and even more prominently in old age. They highlight oligodendroglia of dorsal origin as superior in terms of their remyelinating properties, suggesting that these may be a more promising target for remyelinating treatments.

### 3.2. Regional Heterogeneity

Oligodendroglia also show prominent heterogeneity based on their regional identity in white or grey matter (GM). OPCs show regional differences in their intrinsic propensity for self-renewal, as Hill et al. [32] demonstrated that in response to PDGF-AA signalling, mouse OPCs from WM/GM show different rates of proliferation, with WM OPCs proliferating at consistently higher rates. Crucially, they showed that this is an intrinsic feature of oligodendroglia, as proliferation rates were not affected by transplantation into the antithetical tissue, and the effect was consistent across fore- and hindbrain areas. Similar differences between GM and WM are also found in terms of the ability of oligodendroglia to differentiate. Vigano et al. [31] showed that this is an intrinsic property, as OPCs from the WM consistently gave rise to more mature oligodendrocytes than those from GM, also when transplanted into the heterotopic environment. These studies demonstrate important differences in the heterogeneity of the intrinsic properties of white and grey matter oligodendroglia, showing that the propensity to proliferate and differentiate are both greater in the WM.

Oligodendroglia also show regional heterogeneity between different areas of the CNS, a prominent example being the difference in myelin sheath lengths produced by oligodendrocytes. It has long been recognised that in the CNS, oligodendrocytes in the spinal cord (SC) produce longer myelin sheaths than those produced by cortical oligodendrocytes [29,40], and more recently Bechler et al. [34] demonstrated that this is an intrinsic property of oligodendrocytes, that is independent of axonal cues. In this study, oligodendroglia were cultured on polymer microfibres so as to eliminate any effect of axonal signalling, and showed that SC oligodendrocytes generated longer sheaths than cortical oligodendrocytes on all diameter fibres tested, demonstrating that intrinsic differences between oligodendrocyte populations lead to differences in myelin sheath length.

## 4. Oligodendroglial Heterogeneity at Single-Cell Resolution

While some aspects of oligodendroglial heterogeneity, such as their morphology [29], have been well established and characterised over the past century, others have come to light only in recent years with the advent of novel technologies to investigate cellular diversity at single-cell or single-nuclear resolution [37,41]. The development of droplet-based single-cell RNA sequencing (scRNAseq) has enabled high-throughput transcriptomic profiling of individual cells and nuclei in the CNS, through the barcoding of single RNA molecules in individual cells or nuclei [42,43]. This has become an important tool for researchers to study the heterogeneity of many cells, including neural cell populations such as oligodendroglia (reviewed by van Bruggen et al., 2017 [44]), and to understand the contributions of different cell types and states to disease pathology. There is now increasing evidence to suggest that the highly heterogeneous nature of oligodendroglia might underpin their role in a range of other neurological disorders where their role is less well understood, such as MDD, AD, and PD.

The first investigation of the adult nervous system using high-throughput single-cell transcriptomics was carried out in mouse by Zeisel et al. [45]. They identified prominent oligodendroglial heterogeneity in the mouse cortex and hippocampus, and reported six oligodendrocyte subclusters representing different stages of maturity. In a subsequent study, Marques et al. [37] focused in on the transcriptional heterogeneity of oligodendroglia across ten different regions of the juvenile and adult mouse CNS. They identified twelve subclusters made up of OPCs, committed OPCs (COPs), newly formed oligodendrocytes (NFOL), myelin forming oligodendrocytes (MFOLs) and mature oligodendrocytes (MOLs), and demonstrated that these subpopulations were regionally heterogeneous and represented different states along a continuum of differentiation and maturation. Additionally, they were able to identify a cluster of Itpr2-expressing oligodendroglia involved in the process of activity-dependent myelination, providing evidence that the observed transcriptional heterogeneity in the mouse oligodendroglial lineage is of functional relevance. Activity-dependent or adaptive myelination refers to changes in oligodendrocytes and myelination that occur in response to neuronal activity, resulting in the strengthening of salient pathways. Due to their capacity for adaptive myelination, the identified cluster of Itpr2+ oligodendroglia might be particularly important for remyelination in the context of disease.

The same high-throughput approach has also been used for transcriptional profiling of the human CNS at single-cell resolution, using nuclei extracted from PM donor tissue [6,7,10,11,37,38,40,41,46,47]. Lake et al. [38] carried out transcriptomic and epigenetic profiling of 60,000 nuclei in human PM brains sampled from the visual and frontal cortex, and lateral cerebellar hemispheres of 6 healthy donors. They identified prominent transcriptional heterogeneity in the human CNS and identified 35 clusters, including all major neuronal and glial subtypes, as well as certain specialised cell types such as cerebellar Purkinje cells. They further demonstrated regional heterogeneity of OPCs, with one OPC population identified as specific to the cerebellum. They also compared both single-nuclear and single-cell data from the same region in humans and mice and generated largely consistent cell-type classifications, demonstrating that sequencing of nuclei from the CNS yields largely comparable results as when using whole cells.

The first larger-scale investigation of oligodendroglial heterogeneity in the human brain using single-nuclear transcriptomics was conducted by Jäkel et al. [41], who demonstrated that oligodendroglia isolated from WM in the human PM brain form heterogeneous subpopulations and identified nine distinct clusters of oligodendroglia. As in the murine brain, these were shown to differ along a continuum of maturation, by the ordering of oligodendroglial states along a pseudotime trajectory. Furthermore, the results suggest that this heterogeneity of human oligodendroglial populations has functional consequences, as certain clusters of oligodendroglia express the transcriptional machinery necessary for myelination, whereas others express transcripts associated with cell signalling and viability. Furthermore, an important aim of this study was to investigate the heterogeneity of oligodendroglia in MS, a neurological disease characterised by demyelination, by comparing the transcriptomic profiles of nuclei from the brains of healthy donors and patients with MS. The results showed that a number of oligodendroglial subpopulations were either enriched or depleted in MS, demonstrating that oligodendroglial heterogeneity is altered in this disease as compared to the healthy brain. This has prompted the question of whether changes in the transcriptional heterogeneity of oligodendroglia may be a feature of other neurological disorders where their role is less well understood, such as neuropsychiatric disorders like AD, MDD and PD.

### 4.1. Major Depressive Disorder

Nagy et al. [10] carried out an investigation of transcriptomic profiles in the PM brain of patients with MDD, investigating the cellular heterogeneity of the dorsolateral PFC, a region implicated in this condition. In 80,000 nuclei from 34 male donors, they identified five oligodendroglial subclusters (2 OPCs, 3 oligodendrocytes) and computed their correlation with the clusters identified by Jäkel et al. [41]. Both OPC clusters showed an excellent correlation in terms of their transcriptomic profile with the OPC cluster identified by Jäkel et al. [41], whereas the correlation between oligodendrocyte clusters ranged from r = 0–0.48. Similar to previous studies in both mouse [37] and human [41], the oligodendroglial clusters could be ordered along a pseudotime differentiation trajectory. Furthermore, two of the oligodendroglial clusters identified (OPC2, Oligo1) were depleted in patients relative to controls, providing another example of altered oligodendroglial heterogeneity in a neuropsychiatric condition.

This study also interrogated the differential gene expression of single cells in MDD, in the context of previously published genome-wide association study (GWAS) results. Three of the genes that were differentially expressed between MDD patients and controls had been identified as disease-relevant in previous GWASs of MDD. One of these genes, *KAZN*, was upregulated in the cluster of OPCs enriched in MDD. Crucially, a single-nucleotide polymorphism in this gene has previously shown one of the strongest genome-wide associations with treatment resistant depression [48].

### 4.2. Alzheimer’s Disease

Using similar techniques, Mathys et al. [6] carried out an investigation of transcriptomic profiles in the PM brain of patients with AD. Using droplet-based single nuclei RNA sequencing (snRNAseq) of 80,000 nuclei isolated from the PFC (BA10) of 48 healthy and diseased donors, they identified eight oligodendroglial subpopulations (5 oligodendrocyte, 3 OPCs). Of these, one population of oligodendrocytes and one of OPCs were enriched in AD patients relative to controls, and another oligodendroglial population associated specifically with the absence of pathology. Furthermore, they computed the correlation between neuropathological traits and gene expression, and found an association between AD pathology and the transcriptional pathway for oligodendrocyte differentiation, suggesting a dysregulation of this pathway in the disease. Dysregulation of genes that regulate myelination was observed across the oligodendroglial lineage, suggesting a fundamental impairment to this process in AD.

### 4.3. Parkinson’s Disease

A recent study by Agarwal et al. [7] has established a likely role of oligodendroglia in PD by investigating the transcriptomic profiles of single nuclei from PM substantia nigra (SN) and cortex of patients with PD and controls. One of the key findings in this study is that oligodendroglia in the SN were associated with the disease, which is surprising as the primary pathology of PD is the loss of dopaminergic neurons in this area, and these are only sparsely myelinated. This could suggest that the role of oligodendroglial dysfunction in PD is not through their role as myelinating cells, but that some other aspect of oligodendroglial interaction with the dopaminergic neurons is impaired. Furthermore, the authors observed that the expression of *LRRK2*, a gene that is known to cause a genetic form of PD, was higher in OPCs than any other SN cell type. This adds further evidence to the hypothesis that oligodendroglia may have an important role in mediating non-cell autonomous neurodegeneration in PD and perhaps other neurological/neuropsychiatric diseases.

Using available GWAS data, the authors of this study [7] also compared the cell-type specific transcriptomic expression patterns to the previously identified risk genes for PD, in order to identify disease-relevant cell types. Interestingly, in this cell-type association analysis of the SN they showed association between the genetic risk of PD and the transcriptomic profiles of oligodendroglia. They confirmed that this association was independent of the association between PD risk genes and dopaminergic neurons, providing, for the first time, evidence that there may be distinct disease-associated aetiologies within the SN. The authors also applied this approach in the context of other neuropsychiatric disorders, and in the SN they found an association between genetic risk of Sz and both oligodendrocyte and OPC transcriptomic profiles. Finally, they investigated whether a shared signature of genetic risk could be observed between the different neuropsychiatric conditions. Crucially, they found the same risk genes upregulated in oligodendroglia in both PD and Sz, whereas in neurons different risk genes were upregulated in the two conditions. This provides some tentative evidence of the possibility that there is an oligodendroglial phenotype common across different disease states, similar, e.g., to disease-associated microglia [49]. In order to gain a more detailed understanding of this, an important future direction is to make use of the rapidly growing pool of publicly available snRNAseq data from different neuropsychiatric conditions in order to integrate these and look for commonalities and differences in the transcriptomic profiles of oligodendroglia across different conditions. This will be crucial in determining whether a universal, disease-associated oligodendroglial phenotype or phenotypes exist.

As exemplified by the studies detailed above, another important avenue for future work to elucidate the role of oligodendroglial heterogeneity in neuropsychiatric disease is to utilise single-nuclear transcriptomics alongside other data in order to triangulate on important pathways and mechanisms in a cell-type specific way, e.g., using GWAS data. GWAS has become a widely used approach to generate hypotheses about genes which enhance the risk of diseases including neuropsychiatric conditions, but there are a number of limitations that complicate their interpretation. Firstly, the vast majority of genetic variants identified by GWAS are located in non-coding regions [50] and determining causality of variants is often precluded by the extent of linkage disequilibrium in the human genome. Secondly, by collecting DNA from, e.g., blood or saliva, we do not gain any information on the role of genome-wide significant genetic variants in the CNS or its diverse cell types. By combining GWAS data with single-cell/nuclear transcriptomics, we can begin to elucidate which genetic variants identified by GWAS are disrupted in specific cell types of the CNS, and start to unpick the biological pathways relevant for disease in a cell-type specific manner.

## 5. Future Directions

While still a fairly novel technique subject to rapid development, the use of droplet-based high-throughput snRNAseq sequencing has already provided important insights into the cellular heterogeneity of the human CNS. The use of this method to study human oligodendroglia has highlighted their likely involvement in neurological and neuropsychiatric diseases [6,7,10,41], and provides some basis for speculation on whether the changes in oligodendroglia are specific to each disease state or shared across a number of conditions. Going forward, the ongoing development of novel applications for single-cell/nuclear transcriptomics, including spatially resolved methods and multiomic approaches that combine transcriptomics with, e.g., proteomics and epigenomics, will enable a more detailed understanding of the role of oligodendroglia in neuropsychiatric disease. This may in turn pave the way for developing novel treatments for neuropsychiatric conditions by targeting oligodendroglial dysfunction. Oligodendroglia represent a more tractable target for treatment, as they can be replaced, e.g., from proliferative OPCs. Therefore, they are more manipulable than neurons, and a therapeutic strategy may be to improve clinical phenotypes by targeting oligodendrocytes to improve their own function with a secondary effect on the surrounding neurons, regardless of whether the primary cause of the disease affects the oligodendroglia or the neurons.

## Figures and Tables

**Figure 1 life-11-00125-f001:**
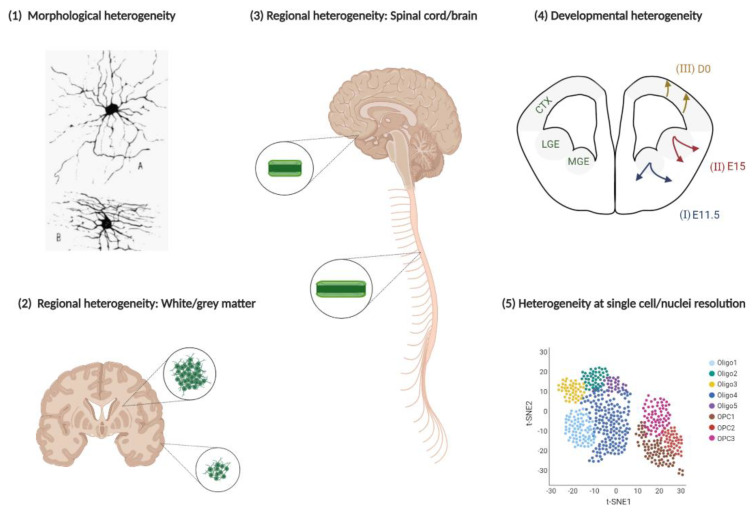
Different types of oligodendroglial heterogeneity in the CNS. **(1)** Oligodendroglia display diverse morphologies, as initially described by Rio del Hortega [29]. (**2**) Oligodendroglia in the white matter show a greater intrinsic propensity to proliferate [32] and differentiate [31], compared with those from grey matter. (**3**) Oligodendroglia in the spinal cord show an intrinsic ability to produce longer myelin sheaths than those in the brain [34]. (**4**) Oligodendroglia develop in three distinct waves, where the two earliest waves of oligodendroglia of ventral origin (I and II) are outcompeted by the later-born dorsal oligodendroglia (III) [36], and the different developmental origins have functional relevance, e.g., for the process of remyelination [35]. (**5**) A number of recent investigations have shown that oligodendroglia are highly heterogeneous in the mouse [37] and human [38] CNS, at single-cell and nuclear resolution respectively.

## Data Availability

Not applicable.

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
