# Peer review of "Oligodendroglial Heterogeneity in Neuropsychiatric Disease"

_life, 2021, doi:10.3390/life11020125_

Round 1

Reviewer 1 Report

This timely review provides an overview of the potential roles that oligodendroglial and myelin changes might play in neuropsychatric diseases, with a particular focus on how oligodendroglial heterogeneity plays into that. The manuscript is well written and organised and cites relevant clinical and experimental data.

Minor comments

In the abstract, it is stated that "Oligodendroglia (defined later as the entire lineage), interact with neurons to support their health....". Suggest adding a short description of the ways oligodendroglia support neurons and neuronal function, in the introduction. This, to help the reader understand how changes in oligodendroglia and myelin might impact neurological function, contributing to neuropsychiatric dysfunction.

Suggest citing evidence that primary oligodendrocyte changes can cause secondary neuronal injury/dysfunction, to support the suggestion ..... "to develop treatments". 

Suggest introducing the concepts of "altered oligodendroglia heterogeneity in disease" (lines 254-60) in the introduction.

Line 27 suggest "neurological" versus "behavioural"

Line 28 consider "myelin changes" to encompass hypo-, dys- and demyelination

Suggest defining, in the introduction, the various ways in which oligodendroglial/myelin changes have been evaluated in neuropsychiatric disease  e.g. transcriptomic profiling of post-mortem material, MRI (including DTI), histological evaluation of post-mortem material.

Line 94 "Gene expression" perhaps

Line 101 "diffusivity" refers to radial diffusivity, presumably; as assessed using DTI

Lines 265-267 "Of these, one population of oligodendrocytes and one of OPCs were enriched in AD patients relative to controls, with a different oligodendroglial population associated specifically with the absence of pathology" It is a unclear what "a different ...population" refers to.

Line 269 "was found" to be deleted

Suggest italicise gene symbols e.g. KAZN on line 289 and LRRK2 on line 301, and throughout. 

Lines 314-315 "..found an association between genetic risk of Sz and both oligodendrocytes and OPCs". This sentence is a little unclear. OL and OPC transcriptomic profiles, perhaps?

Reviewer 2 Report

I propose you to to change the title of this article, because Parkinson, Huntington and Alzheimer are Neurogenerative diseases and not psychiatric disorders

1. "Oligodendroglial Heterogeneity in Neuropsychiatric Disease": "Oligodendroglial Heterogeneity in Neurodegenerative and Neuropsychiatric Disease"

2.Add the abbreviations in the front of the article

3. 89: The role of oligodendroglia in other neuropsychiatric diseases 

I propose you to suppress "other" 

4. 259:;;;orders where their role is less well understood, such as neuropsychiatric disorders like AD, MDD and PD.

such as "neurodegenerative disorders like AD, PD and neuropsychiatric disorders like MDD and Sz.

5. change the order of the paragraphs: 4.1. Alzheimer’s Disease; 4.2. Parkinson’s Disease and Sz; 4.3. Depressive Disorder.

6. 321: I suggest you to reformulate the conclusions
